# Prevalence and Molecular Epidemiology of Extended-Spectrum-β-Lactamase (ESBL)-Producing *Escherichia coli* from Multiple Sectors of Poultry Industry in Korea

**DOI:** 10.3390/antibiotics10091050

**Published:** 2021-08-28

**Authors:** Hyunsoo Kim, Young Ah Kim, Young Hee Seo, Hyukmin Lee, Kyungwon Lee

**Affiliations:** 1Department of Laboratory Medicine, National Police Hospital, Seoul 05715, Korea; comichaha@gmail.com; 2Department of Laboratory Medicine, National Health Insurance Service Ilsan Hospital, Goyang 10444, Korea; 3Research Institute of Bacterial Resistance, Yonsei University College of Medicine, Seoul 03722, Korea; heeyae21@daum.net (Y.H.S.); HMLEE71@yuhs.ac (H.L.); LEEKCP@yuhs.ac (K.L.); 4Department of Laboratory Medicine, Yonsei University College of Medicine, Seoul 03722, Korea

**Keywords:** extended-spectrum-β-lactamase, *Escherichia coli*, poultry, core gene multi locus sequence typing

## Abstract

The aim of this study was to investigate the molecular epidemiology of extended-spectrum-β-lactamase producing *Escherichia coli* (ESBL-EC) from poultry, the poultry farm environment, and workers in Korea. A total of 1376 non-duplicate samples were collected from 21 poultry farms, 20 retail stores, 6 slaughterhouses, and 111 workers in a nationwide study in Korea from January 2019 to August 2019. The overall positive rate of ESBL-EC was 6.8%, with variable positive rates according to sources (0.9% of worker, 5.2% of poultry, 10.0% of chicken meat, and 14.3% of environment). Common ESBL types were CTX-M-55 and CTX-M-14 in a total of 93 ESBL-EC isolates. Whole genome sequencing revealed that 84 ESBL-EC isolates had an outstanding accumulation of numerous antimicrobial resistance (AMR) genes associated with resistance to various classes of antimicrobials for human use and well-known antimicrobial gene (ARG)-carrying plasmids. Core gene multi locus sequence typing, using 2390 core genes, indicated no dominant clone or common type in each province. In conclusion, the isolation rates of ESBL-EC were not negligible in the poultry industry-related samples, sharing common ESBL types of human ESBL-EC isolates in Korea.

## 1. Introduction

Bacteria from livestock may carry clinically relevant resistance genes associated with resistance to veterinary and human antimicrobials [1]. Exposure to multidrug-resistant bacteria during farming, slaughtering, and distribution can serve as a vector of transmission to occupationally exposed workers [1]. To establish an effective strategy for the control of AMR, it is very important to understand the current epidemiology and transmission mode of extended-spectrum-β-lactamase-producing *Escherichia coli* (ESBL-EC) among humans, food animals, retail meat, and environments.

*E. coli* is now the most frequently isolated ESBL-carrier, and CTX-Ms have become the most frequently isolated ESBLs [2]. CTX-M β-lactamases were first reported in Japan in 1986 [3]. Since 2000, CTX-M β-lactamases have been reported increasingly in both humans and animals and are now the dominant type of ESBL, replacing classic TEM- and SHV-type ESBLs in most parts of the world [4]. Currently, there are more than 120 different CTX-M β-lactamases clustered in five groups (CTX-M-1, 2, 8, 9, and 25) [5]. Among the CTX-M-type enzymes, CTX-M-15 and CTX-M-14 have repeatedly been reported in most areas of the world, including in South Korea [6].

ESBL-EC has been isolated from cloacal and boot swabs in farms [7], wastewater and process water from slaughterhouses [8], and commercialized chicken carcasses [9]. However, information about the current epidemiology is very limited in South Korea except for a few reports about ESBL-EC in the commercial layer of chicken farms [10] and poultry slaughterhouses [11].

Thus, the purpose of this study was to evaluate the current prevalence and molecular epidemiology of ESBL-EC in multiple sectors of the poultry-related industry in Korea. Data about the transmission of antimicrobial resistance mechanisms among three sectors (human-animal-environment) could be used to introduce a monitoring system for surveillance and develop an effective strategy for the control of AMR.

## 2. Results

The overall positive rate of ESBL-EC was 6.8% (93/1376). It varied according to sources (0.9% in workers, 5.2% in poultry, 10.0% in chicken meat, and 14.3% in environment) and provinces (7.7% in Seoul/Gyeonggi, 7.3% in Gangwon, 4.5% in Jeolla, 6.6% in Chungcheong, and 7.5% in Gyeongsang) (Table 1). ESBL-EC isolates showed high resistance rates to nalidixic acid (84.8%), ciprofloxacin (69.6%), tetracycline (58.4%), chloramphenicol (56.2%), cotrimoxazole (34.5%), and gentamicin (21.3%). Fortunately, all isolates were susceptible to imipenem, meropenem, ertapenem, and colistin (Appendix A).

Most ESBL-EC produced CTX-M type ESBLs (N = 93) without multiple ESBLs. Common ESBL types were CTX-M-55 (N = 33) and CTX-M-14 (N = 33), followed by CTX-M-1 (N = 11), CTX-M-15 (N = 7), CTX-M-65 (N = 5), CTX-M-27 (N = 3), and CTX-M-61 (N = 1) (Table 2). The relative proportion of ESBL genotypes showed differences among poultry industry-related samples depending on the sources (Table 2). Only one worker from a slaughterhouse was colonized with CTX-M-14-producing *E. coli*.

Whole genome sequencing (WGS) of available ESBL-EC isolates (N = 84) revealed that many antimicrobials resistance (AMR) genes associated with resistance to various classes of antimicrobials for human use were detected, such as *bla*, *str*, *aad*, *aac*, *aph*, *mph*, *aac(6′)Ib-cr*, *Qnr*, *dfr*, *sul*, *tet*, *cat*, *fos*, *ARR-3*, and *Inu* (Appendix A). Various plasmid types (IncB/O/K/Z, IncFIA, IncFIB, IncFIC, IncFII, IncHI2, IncHI2A, IncI1, IncI2, IncN, IncQ1, IncX1, IncY, p0111, p0112, and an additional set of Col plasmids) were detected (Appendix A).

ST 93 is the most common (17.9%) with many heterogeneous sequence types (STs). The worldwide pandemic clone, ST131 *E. coli*, was detected from only one sample of the farm environment (Appendix A). Statistically, the distribution of STs were different by provinces, but not by sources (Appendix A). Pairwise post-hoc analysis showed significant differences in the distribution of STs between Seoul/Gyeonggi and Gangwon (*p*-value < 0.0001, Appendix A). Core gene multi locus sequence typing (cgMLST), using 2390 core genes, indicated no dominant clone or common type in each province, meaning that most of the ESBL-EC isolates were not associated with each other (Appendix A).

## 3. Discussion

The resistance network among environmental, community, and nosocomial superbugs has been suggested [1]. The widespread use of antibiotics has resulted in the generation of antibiotic concentration gradients in humans and livestock. It has accelerated the emergence and spread of antibiotic-resistant bacteria among humans and animals [12]. Antibiotic use in food-producing animals can cause the selection of antibiotic-resistant bacteria in animals, contaminate surrounding environments, and change the environment microbiome, which may serve as a resistance pool [13].

In this study, ESBL-EC isolation from the swine industry-related samples was not negligible, with an overall positive rate of 6.8%. Especially, positive rates of ESBL-EC isolation from chicken meat samples were as high as 10.0%. This is worrisome in aspects of food safety. Many poultry-related ESBL-EC can produce CTX-M-14 and CTX-M-15 commonly detected in clinical samples [14,15].

We could not detect carbapenemase-producers and colsitisn-resistant *E. coli* in this study. The spread of carbapenemase-producing enterobacteriaceae (CPE) from food-animals or the environment to humans is also an essential concern of public health. Recently, NDM-5-producing *E. coli* isolated from a dog and a cat in Korea has been reported [16]. However, the information about the isolation of the *mcr* gene from food animal-originated *E. coli* is still limited in Korea [17,18].

Plasmids as diverse and self-replicating extrachromosomal elements and specific plasmid types have been associated with virulence and/or AMR [19]. Interestingly, IncB/O/K/Z, IncFIA, IncFII, and IncQ1 plasmids known to carry carbapenamase genes were detected in this study. For example, an *E. coli* isolate carrying an IncB/O/K/Z-like plasmid encoding a *bla*_CMY-2_ gene confers resistance to carbapenems with an overexpression of CMY-2 and high plasmid copies [20]. IncFIA plasmid-carrying *bla*_NDM-1_ among *Klebsiella pneumoniae* and *Enterobacter cloacae* isolates [21], and IncFII plasmid encoding NDM-1 from ST131 *E. coli* have been reported [22]. IncQ1 plasmids harbor *bla*_KPC-2_ in high-risk lineages of *K. pneumoniae* CG258 [23]. Considering the spread of these plasmids in poultry-related ESBL-EC, continuous monitoring of CPE from food-animals or the environment is essential.

In this study, many ARGs associated with resistance to various classes of antimicrobials, such as β-lactam (*bla*), streptomycin (*str*), aminoglycoside (*aad*, *aac*, *aph*), macrolide (*mph*), quinolone (*aac(6′)Ib-cr*, *Qnr*), trimethoprim-sulfomethoxazole (*dfr*, *sul*), tetracycline (*tet*), chloramphenicol (*cat*), fosfomycin (*fos*), rifampin (*ARR-3*), limcomycin (*Inu*), and multidrug efflux pump (*floR*, *oqxA*, *oqxB*) were detected. The use of antimicrobials identical or belonging to the same classes of importance to humans for food-producing animals could be a serious problem. It has been reported that the largest volume of antimicrobials sold is for pigs (48–57%), followed by that for poultry (18–24%), fisheries (11–25%), and cattle (5–8%) in Korea during 2003–2012 [24]. Tetracycline was the highest selling antimicrobial, followed by penicillins and sulfonamides [24].

The cgMLST indicated no dominant clone or common type in each province, meaning that most ESBL-EC isolates were not associated with each other. This suggests that there is no definite evidence about the emergence of animal-specific clones like livestock-associated methicillin-resistant *Staphylococcus aureus* [25]. In this study, clinically common genotypes of ESBL were prevalent in poultry and chicken meat samples. Thus, an indirect contact transmission between humans and chickens should be considered.

## 4. Materials and Methods

### 4.1. Study Design, Duration, Sampling and Isolation

We collected 1376 non-duplicate samples from 21 poultry farms, 20 retail stores, 6 slaughterhouses, and 111 workers in a nationwide study in Korea from January 2019 to August 2019. These samples included swabs from the following: nose, groin, axillary, antecubital fossa, inter-finger spaces, and stool of workers; human pooled stool in a toilet; nose, skin, rectum, and stool of poultry; fence, floor, ventilation fan of poultry farms; knife, cutting board, and floor of slaughterhouse; knife, cutting board, and showcase of retail shop. Swabbed samples were put into transport media (Copan diagnostics, Murrieta, CA, USA). Soil (25 g), wastewater (1 L), slaughtered poultry (10 g), and chicken meat for sale (10 g) were sampled in sterile containers. Samples were stored at 4 °C before analysis and inoculated in media within 24 h of sampling.

The soil was mixed with 225 mL of buffered peptone water (BD Biosciences, San Jose, CA, USA). The wastewater was filtered with a 0.2 μm filter. The content on the surface of the filter was inoculated into 10 mL of Mueller-Hinton (MH) broth. The slaughtered poultry and chicken meat were inoculated into 10 mL of MH broth under sterile conditions. All inoculated broths were incubated at 36 ± 1 °C for 18–24 h. After enrichment, 10–100 μL of the liquid culture was used for inoculation onto ChromeIDESBL (bioMérieux, Marcy l’E’toile, France) and MacConkey agar (Oxoid, Basingstoke, UK). Swabbed samples were directly inoculated on ChromeIDESBL (bioMérieux) and MacConkey agar (Oxoid). All inoculated agars were incubated at 36 ± 1 °C for 18–24 h.

### 4.2. Strain Identification, Colistin Resistance Screening and Phenotypic Detection of Carbapenemases

The identification of *E. coli* was conducted with the MALDI Biotyper (Bruker Daltonik, Bremen, Germany). The disk diffusion method was used for antimicrobial susceptibility. The diameter of the inhibition zone was interpreted according to CLSI criteria [26]. To detect colistin-resistant isolates, test organisms were screened on MH agar (Oxoid, Basingstoke, UK) containing colistin (0, 1, 2, and 4 μg/mL) with an *E. coli* ATCC™25922 strain. If MIC was >2 mg/L, the isolate was regarded as a colistin-resistant organism [26]. The CarbaNP [27] or modified carbapenem inactivation method was applied [28] if the isolates exhibited carbapenem resistance without compatible carbapenemase genes.

### 4.3. Molecular Characterization of ESBL Genes and Whole Genome Sequencing of ESBL-E. coli Isolates

The PCR and sequencing for ESBL genes (*bla*_TEM_, *bla*_SHV_, and *bla*_CTX-M_) (Appendix A) were performed as described in a previous study [29] for the isolates showing resistance to cefotaxime or ceftazidime. We performed WGS for a total of 84 ESBL-EC. Nine ESBL-EC isolates were excluded because of the repeated failure of the subculture. DNAs of freshly sub-cultured isolates were extracted using a GenElute™ Bacterial Genomic DNA Kit (Sigma-Aldrich, St. Louis, MO, USA) and 8 μg of input genomic DNA was used. Entire genomes of ESBL-EC isolates were sequenced using a NextSeq 550 instrument (Illumina, San Diego, CA, USA). Sequences were assembled with Spades (version 3.11.1) and annotated with Prokka (version 1.13.7). Data of MLST, cgMLST, plasmid typing, and resistance genes were obtained from the website of the Center for Genomic Epidemiology, including MLST 2.0, ResFinder 3.2, and PlasmidFinder 2.0 [30].

### 4.4. Statistical Analysis

Categorical variables (source, province, ESBL type) were described using count and percentage of the group from which they were derived. The relative proportion was defined as the percentage of the total. The prevalence or positive rate was derived by comparing the number of samples that had ESBL-EC with the total number of samples studied. It was expressed as a percentage. Multiple samples from the same poultry were calculated as one sample.

The Chi-square test was used for comparative analysis of categorical variables using IBM SPSS Statistics for Windows software version 23.0 (IBM Corp., Armonk, NY, USA). The statistical significance of the results was defined at *p* < 0.05. Post-hoc pairwise Fisher’s exact tests with Bonferroni’s method were performed to identify which provinces and sources differed significantly, when significant differences (*p*-value < 0.05) were detected. Bonferroni’s correction method was applied to correct multiple tests; the corrected significance limit was a *p*-value < 0.005 (0.05/10).

This study was approved by the Institutional Review Board of National Health Insurance Service Ilsan Hospital, Goyang, Korea as required by hospital policy (approval number: NHIMC 2017-07-041).

## 5. Conclusions

This study aimed to investigate the molecular epidemiology of extended-spectrum-β-lactamase producing *Escherichia coli* (ESBL-EC) from poultry, the poultry farm environment, and workers in Korea. The overall positive rate of ESBL-EC was 6.8%, with variable positive rates according to sources (0.9% in workers, 5.2% in poultry, 10.0% in chicken meat, and 14.3% in the environment). Common ESBL types were CTX-M-55 and CTX-M-14 in a total of 93 ESBL-EC isolates. In conclusion, the proportion of ESBL-EC of poultry industry-related samples was not negligible in Korea. Such epidemiologic data could be used to develop evidence-based policies to reduce AMR and improve animal and human health with the ‘one health’ concept.

## Figures and Tables

**Table 1 antibiotics-10-01050-t001:** Positive rates of ESBL-EC from poultry industry-related samples.

Province	Worker (%)	Poultry (%)	Chicken Meat (%)	Environment (%)	Total (%)
Seoul/Gyeonggi	0/18 (0)	9/140 (6.4)	9/100 (9.0)	4/29 (13.8)	22/287 (7.7%)
Gangwon	0/18 (0)	13/200 (6.5)	4/60 (6.7)	5/25 (20.0)	22/303 (7.3%)
Jeolla	1/9 (11.1)	3/160 (1.9)	6/60 (10.0)	1/14 (7.1)	11/243 (4.5%)
Chungcheong	0/51 (0)	8/160 (5.0)	8/60 (13.3)	3/18 (16.7)	19/289 (6.6%)
Gyeongsang	0/15 (0)	10/160 (6.3)	7/60 (11.7)	2/19 (10.5)	19/254 (7.5%)
Total	1/111 (0.9)	43/820 (5.2)	34/340 (10.0)	15/105 (14.3)	93/1376 (6.8%)

Positive rates = numbers of ESBL-EC-isolated samples × 100/numbers of cultured samples. Abbreviation: ESBL-EC, extended-spectrum-β-lactamase-producing *Escherichia coli.*

**Table 2 antibiotics-10-01050-t002:** Relative proportion of PCR-determined genotypes of ESBL-EC isolates from poultry industry-related samples (N = 93).

	Relative Proportion (%)			
	Worker	Poultry	Chicken Meat	Environment
CTX-M-1 group				
CTX-M-1 (*n* = 11)	0 (*n* = 0)	14.0 (*n* = 6)	11.8 (*n* = 4)	6.7 (*n* = 1)
CTX-M-15 (*n* = 7)	0 (*n* = 0)	2.3 (*n* = 1)	14.7 (*n* = 5)	6.7 (*n* = 1)
CTX-M-55 (*n* = 33)	0 (*n* = 0)	34.9 (*n* = 15)	29.4 (*n* = 10)	53.3 (*n* = 8)
CTX-M-61 (*n* = 1)	0 (*n* = 0)	0 (*n* = 0)	2.9 (*n* = 1)	0 (*n* = 0)
CTX-M-9 group				
CTX-M-14 (*n* = 33)	100 (*n* = 1)	39.5 (*n* = 17)	35.3 (*n* = 12)	20 (*n* = 3)
CTX-M-27 (*n* = 3)	0 (*n* = 0)	2.3 (*n* = 1)	2.9 (*n* = 1)	6.7 (*n* = 1)
CTX-M-65 (*n* = 5)	0 (*n* = 0)	7 (*n* = 3)	2.9 (*n* = 1)	6.7 (*n* = 1)
Total	100 (*n* = 1)	100 (*n* = 43)	100 (*n* = 34)	100 (*n* = 15)

Abbreviations: ESBL-EC, extended-spectrum β-lactamase-producing *Escherichia coli*; *n* = number.

## Data Availability

The data would be available upon request.

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
