# Peer review of "Prevalence and Molecular Epidemiology of Extended-Spectrum-β-Lactamase (ESBL)-Producing Escherichia coli from Multiple Sectors of Poultry Industry in Korea"

_antibiotics, 2021, doi:10.3390/antibiotics10091050_

Round 1

Reviewer 1 Report

The authors performed an investigation in Korea about the prevalence and molecular epidemiology of extended-spectrum β-lactamase producing Escherichia coli (ESBL-EC) from poultry, poultry farms, slaughterhouses, meat products, and relative workers. The authors did whole-genome sequencing to detect all the potential antimicrobial resistance genes and plasmids in the isolated strains. Overall, the manuscript is well-designed and performed, providing informative context about the current situation of ESBL-EC in Korea. However, some concerns should be addressed in order to increase the quality of this work.

Major concerns:

  1. The materials and methods should be divided into different subsections, such as Samples; Strain identification; Antimicrobial susceptibility test; Detection of resistant genes; Statistical analysis; and so on.
  2. The authors should briefly introduce the CTX-M types of ESBL in the Introduction. And only one type of CTX-M-14 was detected in one worker. In addition to the mentioned poultry-related CTX-M-14 and CTX-M-15 producing ESBL-ECs, are there any other types that have been reported in humans, either in Korea or other countries? The author should discuss it.
  3. Do lines 145-148 belong to this manuscript? This part may belong to information in the template of the manuscript, should be deleted.
  4. The quality of Figure 2 needs to be improved: the labeling of numbers of the y-axis and x-axis is too small, as well as the legend of the provinces.
  5. Data Availability Statement. If authors have uploaded the data of whole-genome sequencing, the information of the database is suggested to list in this statement. Additionally, the data described in lines 55-58 were not shown in either figure or table.

Minor concerns:

  1. Few abbreviations (e.g., STs in line 88) lack full names and some gene names (e.g., lines 123-124) were not italicized in the manuscript, please go through the manuscript and correct them.
  2. Institutional Review Statement should be listed after Conclusions, not in the Discussion part as described in lines 201-203.
  3. Unit used in the manuscript should be consistent. Line 159, within 24-h of sampling; line 164 and line 168, 18 – 24 hours.
  4. The primers for detecting the ESBL genes in line 177 are better listed or as supplementary table for the reader.
  5. Are the lines 82-87 belong to the footer of Table 3 or the main manuscript, the format should be changed similarly to other paragraphs if belonging to the main manuscript.

Author Response

Thank you very much for reviewing our manuscript and indicating various points for improving the quality of our paper.

The answer to the comment is in the attached file.

Reviewer 2 Report

This study aimed to investigate the molecular epidemiology of ESBL-E. coli from poultry, poultry farm environment, and workers in Korea. The authors found the proportion of ESBL-E. coli of poultry industry-related samples was not negligible in Korea, sharing common ESBL types of human ESBL-E. coli isolates in Korea. Though the authors using “one health” concept to study the antimicrobial resistance (AMR) in the poultry industry in Korea, this manuscript has some problems that need to be revised.

Abstract:

-L25: I do not think “many” is a good word using here. This word is too vague and unscientific.

-L27: “AMR”. Change “ARM gene” into “antimicrobial gene (ARG)”.

Introduction:

-L39: change “antimicrobial resistance” into “AMR”. The same with L51, L73 etc.

Results:

-L54-55: add the statistical analysis result in this sentence.

-Tables: using “three-line table” in all the tables.

-Table 1: add the statistical analysis in the data.

-Table 2: combine the content of Table 2 and Figure 1.

-Table 3: This table is so large and difficult to understand. I suggest to move it to the supplementary materials and just show the ESBL gene types and the carrying plasmid types here.

-L92: “(P-value<0.0001, Table 2)”.

-Figure 2: move it to the supplementary materials.

Discussion:

-L114: using the full spelling of the CPE here.

-L123-127: using italic for the gene names and E. coli.

-L130: use “ARG” to replace the full spelling.

-L144-145: give more explaination for this sentence.

-L145-148: These sentences are very difficult to understand.

Materials and Methods:

-L172: change “Mueller-Hinton” into “MH”.

-L173: ATCCTM25922

-L177: use the italic for the gene.

-L178: change “any isolate” into “the isolates”. Is there any possibility one isolate containing the ESBL gene but does not show the resistance to cefotaxime or ceftazidime?

Author Response

(The authors gave the same response as above.)

Round 2

Reviewer 2 Report

After my reading the manuscript, I have no further comments and am  satisfied with the revised manuscript and respond to my previous comment. So I would like to have the accept suggestion.